# Equivalent Exchange Method for Decision-Making in Case of Alternatives with Incomparable Attributes

**Tatiana Kravchenko** [1] and **Timofey Shevgunov** [2]*

1. Graduate School of Business, HSE University, Myasnitskaya ulitsa 20, 101000 Moscow, Russia
2. Moscow Aviation Institute, Volokolamskoe Shosse 4, 125993 Moscow, Russia
* Correspondence: shevgunov@gmail.com

**Abstract:** The paper is focused on searching for novel methods aimed at improving the performance and usability of a common decision-making process where a panel of experts are assisted by specialized software systems. An equivalent exchange method (EEM) is considered in the paper as a potential candidate for a versatile method applicable in expert decision-making process for solving problems in various subject domains. The method is formally described in the paper in the form of an iterative algorithm where each iteration leads to the reduce in the number of alternatives under consideration until it converges to the preferable one. The key feature of EEM consists in the fact that the original comparison between multiple alternatives described by many attributes measured in different units is replaced by the sequence of simple exchanges between pairs of alternatives where only two attributes are engaged at once. The numerical example illustrating the full run of the algorithm is thoroughly described, so the actions performed in the steps of the algorithm are explained. The case of the successful implementation of EEM as the module of Expert Decision Support System is also presented.

**Keywords:** decision support systems; decision-making; preference matrix; comparison matrix; equivalent exchange method; dominated alternatives; attribute comparison

## 1. Introduction

One of the important elements consisting of mathematical models of a decision-making process is the model describing decision maker's preferences. Such preferences are commonly based on the information personally obtained from a decision maker (DM) or experts. If such information is qualitative and complete in a proper sense, then under certain mathematical assumptions, the DM preferences may be simulated by means of certain value functions and utility functions [1]. In general, the higher the value of the function is, the more preference should be given to the particular alternative [2].

Another approach in decision-making assumes that the value related to some of DM's alternative points of view can be determined by breaking down the description of the whole decision into scalar merit values, which are clearly understandable by the DM and assign unambiguous merit measure as the utility function of each option. Despite the fact that such function is designed preferably so that they are linear, the nonlinear functions may also be engaged due to complicated interaction in the contributions of the attributes describing the alternatives [3].

It is commonplace that the merit of the input information utilized in procedures of a typical decision-making process may be unreliable or contain errors. However, mathematical models based on incomplete, inaccurate or qualitative data, such as data representing opinions obtained from some experts, may generate a possible solution in some cases [4,5]. Even if it appears to be impossible to acquire the information of desired accuracy about weights or conditional importance of a certain set of criteria, utilizing that inaccurate information would still remain possible if some sort of relaxation is allowed, e.g., the extension

of the ranges given to the weights and ranking the criteria by their importance ordering. Thus, the QUERY method based on the research, claiming the general limitations of human ability to process information, principally focuses on a more reliable consecutive input where the DM is involved in some evaluation steps [6,7]. The main drawback of the QUERY method consists in the fact that it can deal only with a traditional multi-criteria choice decision where a small number of alternatives are considered. To overcome this drawback, A.V. Lotov had developed the method so that it remained quite effective in solving decision-making tasks involving many more alternatives [8] than the original method. A similar approach is the Aspiration-level Interactive Method (AIM), which seems to be a useful decision-making tool exploiting the concept of ordered satisfaction which emphasizes the role of criterial trade-offs in a multichoice attribution [9].

The French school headed by Bernard Roy has been especially prolific in developing decision-making tools based upon the concept of outranking. The main idea lying behind their methods is the assumption that each pair of alternatives, marked as A and B, can be compared resulting in a set of weights. These weights reflect the relative importance generated immediately by the DM. Then, the concordance index is introduced so that its value is evaluated via the weights of the criteria chosen for alternative A and alternative B. Then, the discordance index is identified as the output of a function of a special kind. This function is fitted in such a way that its maximum is proportional to the inferiority that alternative A possesses with respect to alternative B over all criteria. The concordance and discordance indices are used together to generate outranking relationships where the user input primarily controls the set of parameters forming all available ranks. This approach is basically intended to assist the DM in shrinking the original list of the alternatives to a significantly shorter one, commonly including alternatives with salient merit measured by different criteria [10]. The basic concept of outranking has been implemented in the PROMETHEE method [11,12] while the stochastic version, further developed, has been realized as ELECTRE family [13]. The further research into methods of ELECTRE family continues to be rather active [14,15]; however, so does the research into PROMETHEE [16]. If the inaccurate or uncertain input data can be represented as probability distributions, a possible approach to solving a problem can be constructed by means of Stochastic Multiobjective Acceptability Analysis (SMAA) [17,18].

Analytic hierarchy process (AHP) of T.L. Saaty [19,20] is widely used by many researchers belonging to different schools all over the world. The DM applying AHP breaks down their decision-making problem into a hierarchy of more comprehensible subtasks, each capable of being analyzed independently. Those hierarchically decomposed elements may relate to all aspects of the general decision-making problem. As soon as the hierarchy has been built, the DM systematically estimates the merits of its various elements with accordance to their influence on the elements which have higher ranks in the hierarchy. During the pair-wise comparison, the DM is supposed to rely on certain data described in the attributed of the compared elements, but usually the DM will use their personal judgments made on basis of the relative significance and importance of elements. Basically, AHP transforms such estimates into numerical values, which may be processed using standard mathematical operation and straightforwardly compared across the whole steps of the task. As a result, a scalar weight, or priority, is set to each element in the hierarchy that provides an opportunity of performing rational and consistent comparison between different and oftentimes incomparable elements. This advantage distinguishes AHP from many other decision-making methods. On the final stage of the process, numerical priorities are evaluated for each of the alternatives being considered in the decision-making task. In spite of the fact that AHP has been met with a lot of criticism [21–24], it still remains very popular with many practitioners.

The overcoming of the well-known downsides of the AHP method can be achieved with applying at least two improvements. The first proposal is exploiting a geometric mean as a more robust replacement for the commonly used eigenvectors of the matrices representing the relative importance of the attributes [22]. The second approach is the REMBRANDT

method that introduces an alternative scaling, which turns out to be more natural in most cases, according to the claim of its authors [25,26]. In cases where several DMs or invited experts are involved, some methodology is required to control the overall process, from the initial idea to a possible implementation. One such example setting up a conceptual and practical tool which helps the various stakeholders to communicate is GUEST methodology [27]. An example where this methodology is successfully implemented in a decision support system for solving the problem of strategic production allocation can be found in [28].

A set of recently designed decision-making methods is presented in [29]. These methods exploit various principles of coordination or alternation of the estimates found simultaneously by the set of criteria. The chosen tasks are related to different problem fields and some of them permit the involvement of several groups of experts sharing their opinions in a unified manner. However, if the DM is not really able to define the right objective function, the comparative analysis of performance indicators can be carried out [30]. This approach with the unified framework showed a promising result in the problem of choosing optimal positions for putting up charging stations supplying electric vehicles [31,32].

The authors of this paper propose the formalization of the witty technique firstly published as one of the ad-hoc operations in [2]. This paper shows that, having been further developed and properly formalized, it can become an efficient decision-making technique. The authors suggested calling it an equivalent exchange method (EEM) since this is the key operation of the method distinguishing it from others. In contrast to the majority of the methods mentioned above, EEM tends to maintain its efficacy under conditions where a partial uncertainty about the target function takes place as well as its ability to make evaluation on both quantitative and ordinal scales, where the value of attributes can be presented correspondingly in a scalar field or in a rank space. For some reason, EEM has not become popular since it was not even listed among the decision-making methods reviewed in the articles [33,34] despite the fact that EEM does not suffer from the downsides that are specific to many decision-making methods listed above [1]. One of the essential features of EEM is an iterative search for the trade-offs between changes in the estimates of alternatives possible among some of their attributes. EEM can be considered as a promising option for practical purposes. However, its fruitful exploitation requires it to be written formally as an algorithm which will be ready for implementation in an appropriate decision-making system deployed as some software solution.

The rest of the paper is organized as follows: Section 2 formally describes the algorithm of EEM with its main diagram, term and basic concepts; all steps carried out at each iteration of the algorithm are disclosed in greater details. In Section 3, a specific example is thoroughly considered where the complete run of the algorithm is simulated. The main ideas lying in the basis of EEM, the case where multiple solutions are obtained and the convergence issue are discussed in Section 4. The paper ends with Conclusions (Section 5), where the current drawbacks of the algorithm are marked and some ways of the future improvement are suggested.

## 2. Materials and Methods

The scheme of the algorithm expressing EEM is drawn in Figure 1. The structure of the algorithm is iterative, so the same sequence of steps will perpetuate until one of the quitting conditions are fulfilled. The conditions that allow quitting the cycle are to be checked after the fourth step of the iteration. However, the cycle is doomed to be finished, and this point is considered in Section 4.5, where the convergence of the algorithm is addressed.

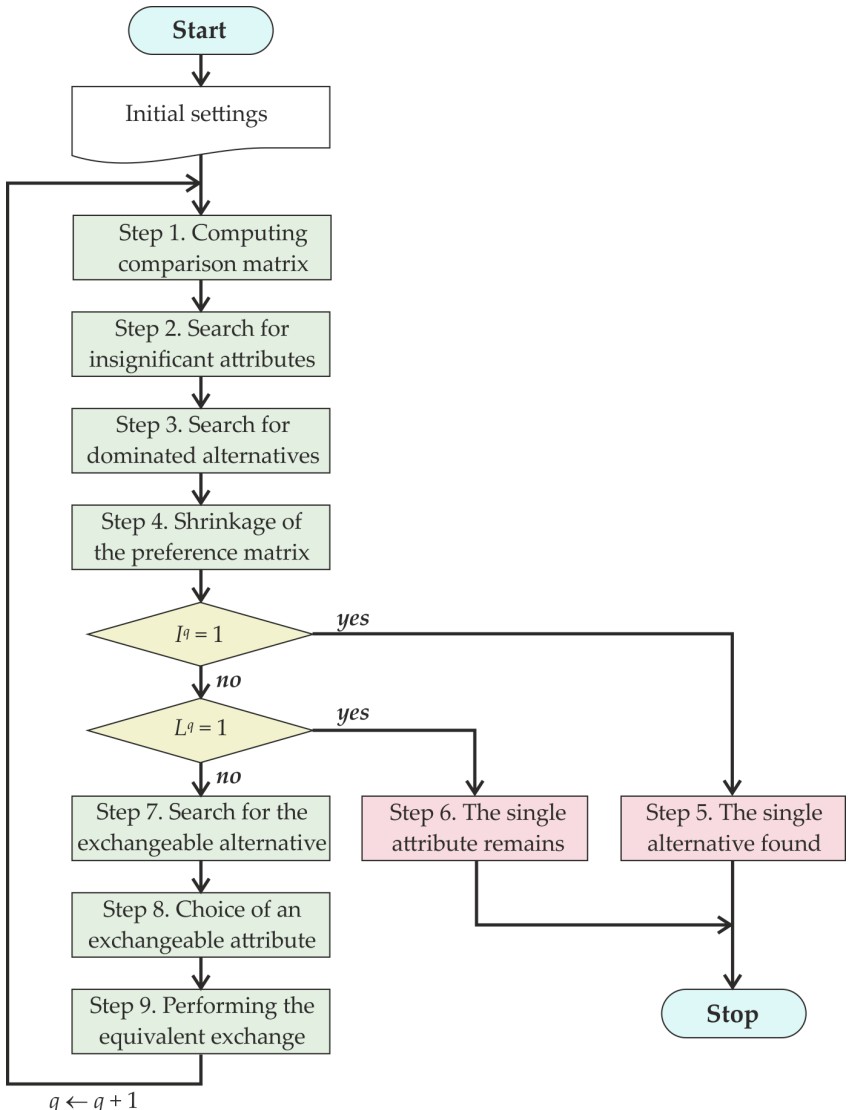

**Figure 1.** The algorithm of the Equivalent Exchange Method.

*2.1. Data Description*

The set of variables used in the algorithm are denoted as follows:

- $q = 1 .. Q$ is the iteration index, where $Q$ is the total number of the iterations, which is written as an upper index in variables.
- $\mathbf{X} = (X_1, X_2, \ldots, X_i, \ldots, X_I), i = 1 .. I$ is the initial set of alternatives, or decision options.
- $I$ denotes the total number of alternatives to be compared.
- $\mathbf{X}^q \subseteq \mathbf{X}$ is the set of alternatives at the $q$th iteration. During the run of the algorithm, the number of alternatives in subsequent iterations is getting reduced, and the alternatives are renumbered.
- $I^q$ marks the number of alternatives at the $q$th iteration.
- $L^q$ designates the number of attributes at the $q$th iteration. In subsequent iterations, the number of attributes cannot increase and tends to go down.
- $l$ denotes the indices of the attributes against which the decision options are compared, $l = 1 .. L^q$.
- $\mathbf{F}^q$ is the preference matrix at the $q$th iteration whose elements $F_{il}^q, i = 1 .. I^q, l = 1 .. L^q$ are estimates of the $i$th alternative by the $l$th attribute. Since the number of alternatives and attributes tend to reduce as the iterations go on, the size of this matrix will also go down.

- $Y_{ikl}^q, l = 1 .. L^q$ stands for the elements of comparison matrices $\mathbf{Y}^q$ for alternative pairs in the $q$th iteration. Each pair of alternatives being compared corresponds to a unique row of the matrix.
- $Y_{ik}^q$ designates the sum of matrix elements $Y_{ikl}^q$ over all the attributes, i.e., the sum of the elements in the row indexed by $(i, k)$.

The usage of triple indexing for elements $Y_{ikl}^q$ requires additional clarification. Thus, the first two indices *ik* form a combined row index, while the the third *l* is the column one. The pairs of alternatives indexing $Y_{ikl}^q$ do not repeat, and any of the pairs cannot contain two identical alternatives. Each of the attributes corresponds to a column of the matrix. Thus, each alternative pair, i.e., the row of the matrix, can be marked by compound index $(i, k), i = 1 .. (I^q - 1), k = (i + 1) .. I^q$. Since neither of the alternative pairs repeats and none of them contains two identical alternatives, the second index *k* will run from $i + 1$ to $I^q$ for each alternative marked with index *i*. Therefore, the total number of rows in the matrix turns out to be equal to the sum of arithmetic progression: $I^q(I^q - 1)/2$.

### 2.2. Basic Concepts

There are two concepts which are important since they are intensively used in the algorithm.

The first concept is dominance which allows for establishing a sort of partial order between the compared alternatives. Basically, the alternative denoted by *i* is called to be *weakly dominating* over the alternative denoted by *k*, i.e., $X_i \succeq X_k$, if $X_i$ is completely not worse than $X_k$. Formally, it means that for all attributes *l* the estimates $F_{il}$ related to the *i*th alternative are not worse than the estimates $F_{kl}$ related to the *k*th alternative:

$$X_i \succeq X_k \iff \forall l, F_{il} \succeq F_{kl}, \tag{1}$$

where the usage of the "greater or equal" sign should be avoided since the bigger value taken on by the attribute does not always imply that it is more preferable. Typically, expenses or ranks may set a good example here: the smaller their values are, the better the related alternative is.

The alternative denoted by *i* is deemed to be *strictly dominating* over the alternative denoted by *k*: $X_i \succ X_k$, if

- $X_i \succeq X_k$, i.e., $X_i$ is weakly dominating over the alternative denoted by $X_k$, AND
- there is at least one attribute $\tilde{l}$ such that the *i*th alternative is better than the *k*th alternative measured with it.

Conversely, in the relations where $X_i \succeq X_k$ and $X_i \succ X_k$, the *k*th alternative can be called dominated, respectfully *weakly dominated* or *strictly dominated*.

The special cases take place in the pair under comparison where the *i*th alternative turns out to be weakly dominating over the *k*th but not strictly dominating. In formal words, $X_i \succeq X_k$ holds when $X_i \succ X_k$ does not. This literally happens if the estimates of both alternatives by all attributes are equal. Then, it is up to the DM to decide which of the alternatives has to be called dominating, and which will be considered as dominated.

The second concept is the insignificance of attributes. The attribute indexed by *l* is called *insignificant* if estimates $F_{il}$ have the same numerical value over all the alternatives $i = 1 .. I^q$ being compared by this attribute. Upon transition to the comparison matrix for alternative pairs, the *l*th attribute is marked as insignificant if the *l*th column of matrix $\mathbf{Y}^q$ is fully populated with 1 as it is shown in the algorithm description below.

### 2.3. The Iteration of the EEM Algorithm

The input data available at the *q*th iteration are the following:

1. The set of alternatives at the *q*th iteration: $\{X_i^q\}, i = 1 \dots I^q$;
2. The set of attributes at the *q*th iteration $l = 1 \dots L^q$;

3.   The preference matrix at the $q$th iteration: $\mathbf{F}^q = \| F_{il}^q \|, i = 1 \ldots I^q, l = 1 \ldots L^q$ is made of the value of preference scalars against the $l$th attribute for alternative $X_i$.

Each iteration of the algorithm consists of nine steps. The operations performed in each step is thoroughly described below accompanied with a numerical example for a better clarification.

### 2.3.1. Step 1: Computing the Comparison Matrix

At the first step, the comparison matrix is built for alternative pairs compared by each of the attributes in accordance with the rule:

$$Y_{ikl}^q = \begin{cases} 1, & F_{il} = F_{kl}; \\ 0, & F_{il} \neq F_{kl}; \end{cases} \quad i = 1 \ldots (I^q - 1), k = (i+1) \ldots I^q. \tag{2}$$

As an example, assume that $I^q = 4$, $\mathbf{X}^q = (X_1, X_2, X_3, X_4)$, $L^q = 5$. Then, for some preference matrix $\mathbf{F}^q$, the corresponding comparison matrix can be evaluated:

$$\mathbf{F}^q = \begin{pmatrix} 30 & 200 & 5 & 16 & 29 \\ 20 & 220 & 5 & 16 & 45 \\ 10 & 200 & 7 & 16 & 35 \\ 10 & 200 & 9 & 16 & 35 \end{pmatrix}, \quad \mathbf{Y}^q = \begin{pmatrix} 0 & 0 & 1 & 1 & 0 \\ 0 & 1 & 0 & 1 & 0 \\ 0 & 1 & 0 & 1 & 0 \\ 0 & 0 & 0 & 1 & 0 \\ 0 & 0 & 0 & 1 & 0 \\ 1 & 1 & 0 & 1 & 1 \end{pmatrix} \begin{matrix} (1,2) \\ (1,3) \\ (1,4) \\ (2,3) \\ (2,4) \\ (3,4) \end{matrix} \tag{3}$$

Each row of matrix $\mathbf{Y}^q$ corresponds to a pair of alternatives being compared $(X_i, X_k)$, which are given to the right for reference. The examples at all other steps of the algorithm will be given for these initial data.

### 2.3.2. Step 2: Search for Insignificant Attributes

As long as the $l$th attribute can be recognized as insignificant, the estimates $F_{il}^q$ for all the alternatives have the same numerical value. Upon the transition to the comparison matrix made for alternative pairs, the $l$th attribute is deemed to be insignificant if the $l$th column of the matrix contains all unity scalar elements. The conditions can be rewritten in a formal manner:

$$\left\{ \forall (i, k), i = 1 \ldots (I^q - 1), k = (i+1) \ldots I^q \mid Y_{ikl}^q = 1 \right\}. \tag{4}$$

In the example started above, the attribute with $l = 4$ turns out insignificant, since the 4th column of matrix $\mathbf{Y}^q$ consists of ones only.

### 2.3.3. Step 3: Search for Dominated Alternatives

The search for dominated alternatives should be carried out among all pairs $(X_i, X_k)$ available at the $q$th iteration. For each of the pairs, the sum of the elements in the corresponding row of the comparison matrix should be computed:

$$Y_{ik}^q = \sum_{l=1}^{L^q} Y_{ikl}^q. \tag{5}$$

This sum basically means the number of attributes, by which the estimates of both alternatives in the current pair are equal. Hereafter, there are three possible cases:

- **Case A** happens if $Y_{ik}^q = L^q$, i.e., the sum of elements in the row of matrix $\mathbf{Y}^q$ corresponding to the pair of alternatives $(X_i, X_k)$ in the $q$th iteration is equal to the number of attributes $L^q$ in the same iteration. This is the case where the weak dominance takes place but not the strict dominance. The DM should make a decision about which of the alternatives is to be kept, and which is declared dominated. The DM may make such a decision straightforwardly or with taking into consideration additional factors, which have not been introduced into the task yet.

- **Case B** happens if $Y_{ik}^q = L^q - 1, L^q > 1$ , i.e., the sum of elements in the row of matrix $\mathbf{Y}^q$ corresponding to the pair of alternatives $(X_i, X_k)$ in the $q$th iteration is less than the number of attributes $L^q$ by 1. Consequently, the estimate of alternatives $(X_i, X_k)$ is equal among all attributes except only one attribute denoted with $\tilde{l}$. In formal words, $\exists! \tilde{l} : Y_{ik\tilde{l}}^q = 0$. The DM must make a decision, which of the estimates by attribute $\tilde{l}$ is preferable: $F_{i\tilde{l}}^q$ or $F_{k\tilde{l}}^q$. If estimate $F_{i\tilde{l}}^q$ is considered preferable, i.e., $F_{i\tilde{l}} \succ F_{k\tilde{l}}$, then alternative $X_i$ should be declared as strictly dominating over alternative $X_k$, $X_i \succ X_k$. Otherwise, alternative $X_k$ would be declared as strictly dominating over alternative $X_i$.
- **Case C** is the situation where neither Case A nor Case B take place. Then, there are no dominated alternatives which have been identified in the step.

It is worth noting that Case C does not imply that there is no dominated alternatives in the set of the alternatives. However, the searching procedure relying on the difference in a single attribute cannot identify it immediately. The more details about this point are discussed in Section 4.6.

In the illustrating example started above, the values of $Y_{ik}^q$ can be found for each pair of alternatives as a sum of all elements of the corresponding rows of matrix $\mathbf{Y}^q$, as it is shown in Table 1.

**Table 1.** The row sums $Y_{ik}^q$ for all alternative pairs.

| Pairs of the Alternatives | | |
|---|---|---|
| i | k | $Y_{ik}^q$ |
| 1 | 2 | 2 |
| 1 | 3 | 2 |
| 1 | 4 | 2 |
| 2 | 3 | 1 |
| 2 | 4 | 1 |
| 3 | 4 | 4 |

In the given example, where $L = 5$, since there are five attributes in $q$th iteration, the indices of the sought alternative pair are $(3, 4)$ as $Y_{34}^q = 4$. In accordance with the conditions, case B takes place, since $Y_{34}^q = L^q - 1$, i.e., alternatives $X_3$ and $X_4$ are equal among all the attributes but $\tilde{l} = 3$. The estimate of alternative $X_3$ by this attribute is $F_{33}^q = 7$, while the estimate of alternative $X_4$ by the same attribute is $F_{43}^q = 9$. The DM must make a decision, which one of the estimates by attribute $\tilde{l} = 3$ is preferable for him or her: 7 or 9. As an example, let us assume that this attribute describes the amount of expenses. Then, estimate $F_{33}^q = 7$ would be preferable. Therefore, alternative $X_3$ would be declared as a dominating one while alternative $X_4$ is to be called dominated.

2.3.4. Step 4: Shrinkage of the Preference Matrix

At this step, all insignificant attributes and dominated alternatives discovered in the previous step (if any) are excluded from consideration:

1.  The dominated alternatives, which are related to both cases A and B in the previous step, have to be excluded from vector $X^q$. The list of remaining alternatives are renumbered to restore the consequent indexing of the alternatives;
2.  The rows corresponding to the dominated alternatives and the columns corresponding to insignificant attributes are deleted from matrix $\mathbf{F}^q$;
3.  The rows in the matrix corresponding to the pairs of alternatives where at least one of the alternatives is being dominated have to be excluded from matrix $\mathbf{Y}^q$. The columns corresponding to insignificant attributes are also excluded;
4.  Variables $I^q$ and $L^q$ take on the new values equal to the number of, respectively, alternatives and attributes remaining for further consideration.

Then, the branching in the run of the algorithm has to be carried out.

If $I^q = 1$, then the execution proceeds with step 5;

If $L^q = 1$, then it proceeds with step 6;

Otherwise, i.e., $(I^q > 1) \wedge (L^q > 1)$, it goes to step 7.

Regarding the example started above, attribute $l = 4$ was identified as insignificant in step 2, while, in step 3, alternative $X_4$ was identified as dominated in step 3. Therefore, alternative $X_4$ is being excluded from vector $\mathbf{X}^q = (X_1, X_2, X_3, \cancel{X_4})$. The remaining alternatives are renumbered so that the vector takes the following form: $\mathbf{X}^q = (X_1, X_2, X_3)$. The transformation made to matrices $\mathbf{F}^q$ and $\mathbf{Y}^q$ can be illustrated as follows:

$$
\mathbf{F}^q = \begin{pmatrix} 30 & 200 & 5 & \cancel{16} & 29 \\ 20 & 220 & 5 & \cancel{16} & 45 \\ 10 & 200 & 7 & \cancel{16} & 35 \\ \cancel{10} & \cancel{200} & \cancel{9} & \cancel{16} & \cancel{35} \end{pmatrix}, \quad
\mathbf{Y}^q = \begin{pmatrix} 0 & 0 & 1 & \cancel{1} & 0 \\ 0 & 1 & 0 & \cancel{1} & 0 \\ \cancel{0} & \cancel{1} & \cancel{0} & \cancel{1} & \cancel{0} \\ 0 & 0 & 0 & \cancel{0} & 0 \\ \cancel{0} & \cancel{0} & \cancel{0} & \cancel{1} & \cancel{0} \\ \cancel{1} & \cancel{1} & \cancel{0} & \cancel{1} & \cancel{1} \end{pmatrix} \begin{matrix} (1,2) \\ (1,3) \\ \cancel{(1,4)} \\ (2,3) \\ \cancel{(2,4)} \\ \cancel{(3,4)} \end{matrix}
\tag{6}
$$

Row 4 has been excluded from matrix $\mathbf{F}^q$, since it corresponds to an alternative $X_4$, and column 4 has also been excluded, since it corresponds to some attribute $l = 3$. The changes in matrix $\mathbf{Y}^q$ are more complicated, since each row corresponds to a pair of alternatives. Thus, all the rows corresponding to the pairs of alternatives, where at least one of the alternatives is a dominated one, are excluded. Since alternative $X_4$ is dominated, the rows indexed by the following pairs have to be excluded: $(X_1, X_4)$, $(X_2, X_4)$, $(X_3, X_4)$. In addition, column 4 is excluded because it describes the attribute found as insignificant. The reduced matrices take on the following values:

$$
\mathbf{F}^q = \begin{pmatrix} 30 & 200 & 5 & 29 \\ 20 & 220 & 5 & 45 \\ 10 & 200 & 7 & 35 \end{pmatrix}, \quad
\mathbf{Y}^q = \begin{pmatrix} 0 & 0 & 1 & 0 \\ 0 & 1 & 0 & 0 \\ 0 & 0 & 0 & 0 \end{pmatrix} \begin{matrix} (1,2) \\ (1,3) \\ (2,3) \end{matrix}
\tag{7}
$$

Variables $I^q$ and $L^q$ take on values equal to the new number of alternatives and attributes, respectively: $I^q = 3$ and $L^q = 4$.

### 2.3.5. Step 5: The Single Alternative Found

If $I^q = 1$, there is only one alternative left in Step 4. Consequently, $X^* = X_1^q$ is the effective alternative.

The algorithm terminates here.

### 2.3.6. Step 6: The Single Attribute Remains

If $L^q = 1$, there only one attribute left in Step 4. Therefore, the DM must choose the alternative with the preferable estimate by the remaining attribute, i.e., to select such alternative $X^* = X_i^q$ that the estimate $F_{i1}^q$ related to the $i$th alternative would be preferable among all. If more than one alternative has the same preferable estimates, then the algorithm output could produce the set made of these alternatives as its output.

The algorithm terminates afterwards.

### 2.3.7. Step 7: Search for the Alternative Suitable for Equivalent Exchange

The goal of this step is the selection of the pair of alternatives indexed as $(\hat{i}, \hat{k})$ and one attribute $\hat{l}$, for which an equivalent exchange is the most suitable to be conducted. In order to perform this, the elements $Y_{ikl}^q$ of $\mathbf{Y}^q$ taking on zero value are considered: Then, the pair of alternatives for which the value:

$$
D_{ik}^q = L^q - \sum_{l=1}^{L^q} Y_{ikl}^q
\tag{8}
$$

reaches the minimum that has to be chosen. Among all elements of matrix $Y_{ikl}^q$ that correspond to such pair of alternatives $(\hat{i}, \hat{k})$, it is required to find such zero element $(\hat{i}, \hat{k}, \hat{l})$ that will satisfy the following condition:

$$\frac{I^q(I^q-1)}{2} - \sum_{i=1}^{I^q-1} \sum_{k=i+1}^{I^q} Y_{ik\hat{l}}^q = \min_l \left\{ \frac{I^q(I^q-1)}{2} - \sum_{i=1}^{I^q-1} \sum_{k=i+1}^{I^q} Y_{ikl}^q \right\}. \tag{9}$$

Due to the fact that the first term of the sum is constant, this condition may be replaced with the following:

$$\sum_{i=1}^{I^q-1} \sum_{k=i+1}^{I^q} Y_{ik\hat{l}}^q = \max_l \sum_{i=1}^{I^q-1} \sum_{k=i+1}^{I^q} Y_{ikl}^q. \tag{10}$$

It is worth noting that, while $D_{ik}^q$ simply indicates the number of zero elements in the row indexed by $(\hat{i}, \hat{k})$, the left-hand side of the expression (9) implicitly expresses the number of zero elements in the column indexed as $\hat{l}$.

In those cases where more than one element $Y_{ikl}^q$ of the matrix that meet this condition has been found, then the DM will have their own right to choose a pair among all identified.

In the continuing example, the elements of the comparison matrix meeting this requirement are underlined:

$$\mathbf{Y}^q = \begin{pmatrix} 0 & \underline{0} & 1 & 0 \\ 0 & 1 & \underline{0} & 0 \\ 0 & \underline{0} & \underline{0} & 0 \end{pmatrix} \tag{11}$$

It is important to notice that each of the underlined elements has two zeros in its column. However, each of the other (not underlined) zero elements has 3 zeros in its column. The DM must choose any of the underlined elements. If they choose element $(\hat{i}, \hat{k}, \hat{l}) = (1, 2, 2)$, for instance, it will be the closest element to the upper left corner of the matrix $\mathbf{Y}^q$ since it is in its row 1 and column 2.

2.3.8. Step 8: Choice of an Exchangeable Attribute

In this step, the DM is set to select such attribute $l^*$ , which the DM considers to be an appropriate candidate for further comparison made in the pair of alternatives chosen in the previous step. Basically, attribute $l^*$ should be chosen to meet two criteria:

1.  Attribute $l^*$ cannot be the same as attribute $\hat{l}$, selected in Step 7;
2.  The element of the comparison matrix that corresponds to this attribute and the pair $(X_{\hat{i}}, X_{\hat{k}})$ selected in step 7 must be zero: $Y_{\hat{i}\hat{k}l^*}^q = 0$.

Provided the previous step are executed correctly, this attribute $l^*$ always exists because otherwise one of the alternatives would be dominated, which means that it should have been deleted in Step 4.

In the previous step of the continuing example, element $(\hat{i}, \hat{k}, \hat{l}) = (1, 2, 2)$ was selected. Therefore, $l^*$ cannot be chosen equal to $\hat{l}$: $l^* \neq \hat{l}$, so the element in the first row and column $l^*$ of matrix $\mathbf{Y}^q$ must be zero. Therefore, both attributes with indices 1 and 4 are suitable. However, this becomes the DM's personal decision: which of them will be chosen as the attribute ready for exchange $l^*$. This choice should be made assuming that the DM will be obliged to compare simultaneous changes made in both of the estimates defined by attributes $\hat{l}$ and $l^*$ in the next step of the algorithm. In the example under consideration, let $l^* = 1$ be assumed.

2.3.9. Step 9: Performing the Equivalent Exchange

In this step, preference matrix $\mathbf{F}^{q+1}$ is formed for the next $(q+1)$th iteration of the algorithm. Initially, the preference matrix of the current iteration is copied into the matrix intended for the next iteration $\mathbf{F}^{q+1} = \mathbf{F}^q$; then, the modification is carried out. Primarily, the DM assigns the new values for the element in $\mathbf{F}^{q+1}$, such that the change in the estimate by attribute $\hat{l}$ from $F_{\hat{k}\hat{l}}^q$ to $F_{\hat{i}\hat{l}}^q$ would be considered as equivalent to the change in the estimate by attribute $l^*$ from $F_{\hat{k}l^*}^q$ to $F_{\hat{k}l^*}^{q+1}$. Since the first exchange can deliberately be assumed as

$F_{\hat{k}\hat{l}}^{q+1} = F_{\hat{i}\hat{l}}^{q}$, the new value of element $F_{\hat{k}l*}^{q+1}$ should be assigned by the DM. Performing these actions, the decrease in the value of

$$M^q = \min_{(i,k)} D_{ik}^q = \min_{(i,k)} \left\{ L^q - \sum_{l=1}^{L^q} Y_{ikl}^q \right\}. \tag{12}$$

should be ensured in the next step, since the minimum is taken for all suitable pairs of alternatives. It follows from the fact that the number of attributes, by which the alternatives index by $(\hat{i}, \hat{k})$ differ, will inevitably be reduced by 1 after the iteration has complete.

Then, here comes the transition to the next iteration, marked as $q + 1$, where the whole procedure is to be repeated starting in Step 1.

Regarding the continuing example, the final step of the current iteration engages the preference matrix (7) obtained in Step 4, triplet $(\hat{i}, \hat{k}, \hat{l}) = (1, 2, 2)$ chosen in Step 7 and attribute $l^* = 1$ selected in Step 8.

In order to make the exchange more convenient and simpler for understanding, let us assume that attribute $l^*$ means "implementation period in weeks", and attribute $\hat{l}$ means "cost in thousand euros". The DM considers it to be reasonable to make the following decision:

*"Let us suppose that the cost of implementation changes from 220 to 200 thousand euros. By how many days should the implementation period be increased from the initial 20 weeks, so that both changes would be equivalent from the DM's own perspective?"*

For example, the DM may reason as follows:

*"To reduce the cost of implementation by 20 thousand euros, we can give 3 extra weeks for system implementation".*

Accordingly, the DM will assign the value $F_{\hat{k}l*}^{q+1} = 23$. After the equivalent exchange has been completed, the preference matrix is formed for the next iteration:

$$\mathbf{F}^{q+1} = \begin{pmatrix} 30 & 200 & 5 & 29 \\ \underline{23} & \underline{200} & 5 & 45 \\ 10 & 200 & 7 & 35 \end{pmatrix}, \tag{13}$$

where the just changed elements are underlined.

This matrix (13) will be used in the next iteration of the algorithm. However, even at the current step, it is rather easy to deduce that attribute 2 is bound to be claimed insignificant since the estimates of all the alternatives by this attribute have become equal to the similar value, namely 200.

### 2.4. The EDSS Description

A formal description of EEM algorithm was developed, which further allowed rewriting it in the terms ready to be implemented as an extending module of the Expert Decision Support System (EDSS) [29]. This information system primarily focuses on automation of the procedures performing analysis of the problem situation where the selection of the effective decision among available ones is required. The specific features of the system are briefly highlighted below.

The EDSS was basically designed for the facilitation of the overall process of making decision. Firstly, the system helps its user, i.e., the DM, to choose a suitable method for particular problem among all available, depending on the amount of the problem related data the DM holds or even personal preferences. Secondly, the system provides the DM with the interactive graphical user interface. Finally, the system conducts all the computations required for running the chosen method.

The knowledge base of the EDSS contains different methods implemented in the form of executable modules. The range of the methods varies from those which are completely based on formal foundation, e.g., optimal solution techniques, to those which rely mostly on the decisions the experts bring in during their sessions. The current version of the

system virtually implements nearly 50 particular instances of such mathematical methods and models, which makes it stand out from most of the other DSS, which usually realize a single decision-making method or a family of similar methods. Among the methods implemented in the system so far, there are also groups of decision-making methods performing under uncertainty and risk, which can be introduced to the simulation while solving a decision-making problem.

As the essential part of the EDSS, its knowledge base additionally contains a set of rules for selection among appropriate models and decision-making methods. The list of suitable methods become available as soon as all elements constituting the task being solved are collected together and properly brought in to the system. The knowledge base assists in the main session, performing the search for the most suitable decision-making methods which are driven by the sequence made of its user's selection of the answer, taken out of the set of locally available answers, to the questions asked by the system about the general elements of the decision-making task. As a result, not only does the system provide a user with a better opportunity to select the decision-making method for a specific problem, but also it guides the user through the solution steps being carried out according to the method.

The EDSS is equipped with a sufficient set of functions to deal with a database server generally supporting SQL via a standard database provider interface. This supports the structured storage of all the information required for internal calculation, the description of task elements, and intermediate results obtained by decision-marking methods. It also stores the predefined templates utilized during the preparation of reports as well as for support of the multi-language user interface. In addition, relying on the database servers, the system can carry out a multidimensional analysis of the data before and during the main procedure of solving the task at hand, export the results back to the database tables, and prepare analytic reports with the use of an OLAP-server. However, the automation based on stored templates will slightly vary the procedure of report generation with respect to the functional specification provided by the server of the chosen relational database.

The system architecture includes a multiple user access which provides the set of tools for collective decision-making process that allows an expert panel to substantiate decisions in a cooperative manner so that the optimal solution can be reached as a consolidated opinion of the expert panel.

The EDSS is not a system oriented exclusively to solving problems in a dedicated field. In spite of the fact that it was primarily designed for solving tasks in managerial [35,36] or business [37,38] sciences, it can now be fruitfully used for coping with tasks emerging in many other applied sciences. As a system improving the process of choosing optimal system design at the earlier development stages, it has served a practical tool for solving a series of engineering tasks, e.g., searching for more effective estimators in radar systems [39,40], selecting more accurate estimators for electric signal parameters [41–43], planning the design of experiments conducting in position location [44], telecommunication [45] and fast image recognition [46], identification of hidden periodicity in data acquired at various measurement sites, and including power electronics [47,48] and antennas [49]. Such approach can also be applied for designing the sites for measurement of the vibration signals exploited for fault detection in rotary machines [50].

Since the system had been designed as a client–server architecture, it provides end users with a remote access to the web server running the main application. The end-user implementation in the form of a thin client offers the simultaneous sessions to multiple users connected via the Internet browser.

The inclusion of the newly developed method into the EDSS has been successfully carried out. The modular architecture of the server part of the system allows for adding new decision-making techniques without changing the source code of the main system modules. However, the user graphical interface of the EDSS has been slightly updated in order to support the interactive function of the included method. Practically, the proposed method has been implemented as a program module written in C# language with a support

of the program interface providing the data structure interchange at a level that is enough for the full inclusion into the system. The knowledge base exploited by the EDSS has also been updated, since the new method had to be implemented in accordance with those decision-making conditions, under which the method is expected to be run.

## 3. Results

For the sake of clarification, the practical execution of the proposed algorithm can be demonstrated by means of an example where the sequence of iterations will be carried out. The case description is as follows. A DM is considering a problem of finding the optimal solution among three alternatives which are the offices for a new branch of a private company.

Five attributes were chosen which quantitatively characterize the essential consumer properties of the office space from the DM's point of view. The first is 'Time' which is measured in minutes and designates the average time which employees are expected to spend in order to get the office from the nearest subway station. Secondly, 'Location' is an expert unitless estimate of the general perception of the quality of the office center and its neighborhood. Then, 'Equipment' is also an expert unitless estimate which determines how much common office equipment has already been placed in the office rooms. The fourth is 'Size' measured in square meters which denotes the area taken by each of the offices. Finally, 'Costs' is the total estimated spending for the rent of the whole office, and 1 unit is equal to €1000 per year.

The values of the attributes are known for three alternatives named as 'Office A', 'Office B' and 'Office C'. All the data collected are used for filling in the initial preference matrix, which is shown in Table 2.

The run of the algorithm will be organized as a sequence of iterations where the preference matrix undergoes gradual modifications—one modification at each step—until it finally comes to the state where the simple scalar comparison is immediately realizable.

**Table 2.** The preference matrix at Iteration 1—$\mathbf{F}^1$.

|  | **1: Time** | **2: Location** | **3: Equipment** | **4: Size** | **5: Costs** |
|---|---|---|---|---|---|
| $X_1$: **Office A** | 25.0 | 80.0 | 60.0 | 700.0 | 1700.0 |
| $X_2$: **Office B** | 28.0 | 70.0 | 80.0 | 500.0 | 1500.0 |
| $X_3$: **Office C** | 25.0 | 85.0 | 50.0 | 800.0 | 1900.0 |

### 3.1. Iteration 1

The preference matrix at the first iteration $\mathbf{F}^1$ is shown in Table 2. The computed pairwise comparison matrix $\mathbf{Y}^1$ is shown in Table 3. As it can be seen from the comparison matrix, there are neither attributes to be called insignificant nor alternatives to be marked as dominated. Therefore, the iteration comes up to Step 7. The choice of the exchangeable alternatives and the attribute $\hat{l}$ for the equivalent exchange is performed by the EDSS itself. Then, in Step 8, the system pushes the dialog window shown in Figure 2. This window inquires the DM after the attribute $l^*$ to perform the equivalent exchange with. The window shows that one of the **Time** attributes for the exchange has been chosen automatically. This selection can be easily explained since there is a unity in the first column of Table 3 while all other columns contained zeros only. This choice will head for a faster elimination of the column as soon as its attribute has been found to be insignificant.

Suppose that the DM has picked up an **Equipment** attribute for the equivalent exchange with **Time**. Then, in Step 9, the dialog window, which is shown in Figure 3, asks the DM for entering the new value of **Equipment** attribute. The estimate of $X_2$ by **Time** attribute is going to decrease from 28 to 25, whiich will be thought of as a positive effect. The DM considers that equivalent decrease in **Equipment** should be its change from 80 to 78. Although this decision may well look arbitrary, it is the simulation of the DM's personal decision here. In other words, the DM is ready to sacrifice just a small piece of

office equipment for a three-minute decrease in time that most of the staff will spend to obtain an office.

The action taken in Step 9 of the first iteration is the equivalent exchange made between $X_1$ and $X_2$ alternatives. The resultant matrix is shown in Table 4.

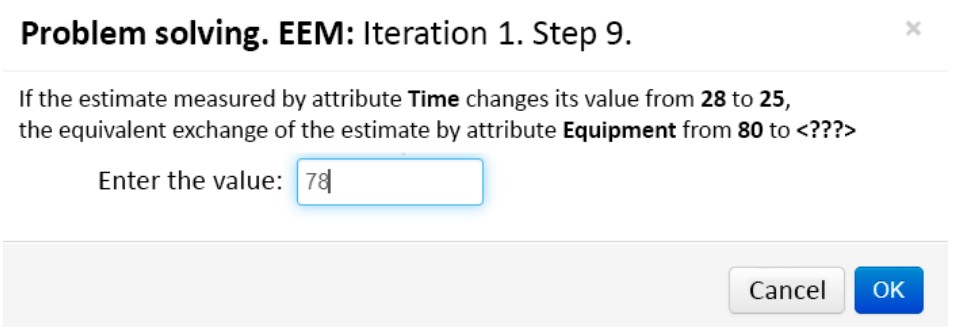

**Figure 2.** The dialog window for choosing the attribute for the equivalent exchange at Iteration 1.

**Problem solving. EEM:** Iteration 1. Step 9.

If the estimate measured by attribute **Time** changes its value from **28** to **25**, the equivalent exchange of the estimate by attribute **Equipment** from **80** to **<???>**

Enter the value: 78

Cancel    OK

**Figure 3.** The dialog window for entering for the equivalent exchange at Iteration 1.

**Table 3.** The pair-wise comparison matrix at Iteration 1—$\mathbf{Y}^1$.

| $(X_i : X_k)/l$ | 1 | 2 | 3 | 4 | 5 | $Y_{ik}^1(\Sigma)$ |
|---|---|---|---|---|---|---|
| $X_1 : X_2$ | 0 | 0 | 0 | 0 | 0 | 0 |
| $X_1 : X_3$ | 1 | 0 | 0 | 0 | 0 | 1 |
| $X_2 : X_3$ | 0 | 0 | 0 | 0 | 0 | 0 |

**Table 4.** The preference matrix at Iteration 2—$\mathbf{F}^2$.

| | 1: Time | 2: Location | 3: Equipment | 4: Size | 5: Costs |
|---|---|---|---|---|---|
| $X_1$: **Office A** | 25.0 | 80.0 | 60.0 | 700.0 | 1700.0 |
| $X_2$: **Office B** | 25.0 | 70.0 | 78.0 | 500.0 | 1500.0 |
| $X_3$: **Office C** | 25.0 | 85.0 | 50.0 | 800.0 | 1900.0 |

*3.2. Iteration 2*

Table 4 demonstrates the preference matrix $\mathbf{F}^2$ at the beginning of the second iteration. The computed pairwise comparison matrix $\mathbf{Y}^2$ is shown in Table 5.

**Table 5.** The pair-wise comparison matrix at Iteration 2—$\mathbf{Y}^2$.

| $(X_i : X_k)/l$ | 1 | 2 | 3 | 4 | 5 | $Y_{ik}^2(\Sigma)$ |
|---|---|---|---|---|---|---|
| $X_1 : X_2$ | 1 | 0 | 0 | 0 | 0 | 1 |
| $X_1 : X_3$ | 1 | 0 | 0 | 0 | 0 | 1 |
| $X_2 : X_3$ | 1 | 0 | 0 | 0 | 0 | 1 |

The unity vector in the first column observes that **Time** is an insignificant attribute and has to be excluded. This exclusion has led to the reduction in the number of columns constituting matrix $\mathbf{F}^2$. The new matrices $\mathbf{F}^2$ and $\mathbf{Y}^2$ are shown in Tables 6 and 7 correspondingly. The number of attributes has decreased, and the rest of the attributes have been renumbered. It is important to emphasize the fact that the insignificance of **Time** and the deletion of its column have become possible because of the equivalent exchange done at the previous iteration—Iteration 1.

Now, the just shrunk comparison matrix shown in Table 7 is searching for the new equivalent exchange. Since there are no unit elements in the matrix at all, the EDSS proposes the attribute $\hat{l}$ as the first in the list, namely **Location**, while the attribute for exchange has to be chosen by the DM. Suppose they have picked up **Costs** and decided that the increase in **Location** from 70 to 80 can reasonably be achieved with an increase in **Costs** from 1500 to 1600. The result of the second iteration is the equivalent exchange made between $X_1$ and $X_2$ alternatives. The resultant matrix is shown in Table 8.

**Table 6.** The preference matrix at Iteration 2 after **Time** has been excluded—$\mathbf{F}^2$.

|  | 1: Location | 2: Equipment | 3: Size | 4: Costs |
|---|---|---|---|---|
| $X_1$: **Office A** | 80.0 | 60.0 | 700.0 | 1700.0 |
| $X_2$: **Office B** | 70.0 | 78.0 | 500.0 | 1500.0 |
| $X_3$: **Office C** | 85.0 | 50.0 | 800.0 | 1900.0 |

**Table 7.** The pair-wise comparison matrix at Iteration 2 after **Time** has been excluded—$\mathbf{Y}^2$.

| $(X_i : X_k)/l$ | 1 | 2 | 3 | 4 | $Y_{ik}^2(\Sigma)$ |
|---|---|---|---|---|---|
| $X_1 : X_2$ | 0 | 0 | 0 | 0 | 0 |
| $X_1 : X_3$ | 0 | 0 | 0 | 0 | 0 |
| $X_2 : X_3$ | 0 | 0 | 0 | 0 | 0 |

**Table 8.** The preference matrix at Iteration 3—$\mathbf{F}^3$.

|  | 1: Location | 2: Equipment | 3: Size | 4: Costs |
|---|---|---|---|---|
| $X_1$: **Office A** | 80.0 | 60.0 | 700.0 | 1700.0 |
| $X_2$: **Office B** | 80.0 | 78.0 | 500.0 | 1600.0 |
| $X_3$: **Office C** | 85.0 | 50.0 | 800.0 | 1900.0 |

*3.3. Iteration 3*

The preference matrix at the third iteration $\mathbf{F}^3$ is depicted in Table 8. The computed pairwise comparison matrix $\mathbf{Y}^3$ is presented in Table 9. Since there is a unit element in the comparison matrix, and it is in the column related to **Location** attribute, the EDSS suggests this attribute as $\hat{l}$ again, while the DM picks up **Size** for the exchange. The decrease in **Location** from 85 to 80 can be tolerated if the equivalent increase in **Location** takes place from 800 to 850.

The third iteration results in the equivalent exchange between $X_2$ and $X_3$ rows. The preference matrix obtained after the third iteration has completed is shown in Table 10.

**Table 9.** The pair-wise comparison matrix at Iteration 3—$\mathbf{Y}^3$.

| $(X_i : X_k)/l$ | 1 | 2 | 3 | 4 | $Y^3_{ik}(\Sigma)$ |
|---|---|---|---|---|---|
| $X_1 : X_2$ | 1 | 0 | 0 | 0 | 1 |
| $X_1 : X_3$ | 0 | 0 | 0 | 0 | 0 |
| $X_2 : X_3$ | 0 | 0 | 0 | 0 | 0 |

**Table 10.** The preference matrix at Iteration 4—$\mathbf{F}^4$.

| | 1: Location | 2: Equipment | 3: Size | 4: Costs |
|---|---|---|---|---|
| $X_1$: **Office A** | 80.0 | 60.0 | 700.0 | 1700.0 |
| $X_2$: **Office B** | 80.0 | 78.0 | 500.0 | 1600.0 |
| $X_3$: **Office C** | 80.0 | 50.0 | 850.0 | 1900.0 |

*3.4. Iteration 4*

Table 10 demonstrates the preference matrix $\mathbf{F}^4$ at the beginning of the fourth iteration. The computed pairwise comparison matrix $\mathbf{Y}^4$ is shown in Table 11. The unity vector in the first column witnesses that **Location** has become an insignificant attribute and has to be excluded. This leads to the reduction in the number of columns constituting matrix $\mathbf{F}^4$ since the corresponding column has to be deleted. The new matrices $\mathbf{F}^4$ and $\mathbf{Y}^4$ are shown in Tables 12 and 13 correspondingly.

**Table 11.** The pair-wise comparison matrix at Iteration 4—$\mathbf{Y}^4$.

| $(X_i : X_k)/l$ | 1 | 2 | 3 | 4 | $Y^4_{ik}(\Sigma)$ |
|---|---|---|---|---|---|
| $X_1 : X_2$ | 1 | 0 | 0 | 0 | 1 |
| $X_1 : X_3$ | 1 | 0 | 0 | 0 | 1 |
| $X_2 : X_3$ | 1 | 0 | 0 | 0 | 1 |

**Table 12.** The preference matrix at Iteration 4 after **Location** has been excluded—$\mathbf{F}^4$.

| | 1: Equipment | 2: Size | 3: Costs |
|---|---|---|---|
| $X_1$: **Office A** | 60.0 | 700.0 | 1700.0 |
| $X_2$: **Office B** | 78.0 | 500.0 | 1600.0 |
| $X_3$: **Office C** | 50.0 | 850.0 | 1900.0 |

**Table 13.** The pair-wise comparison matrix at Iteration 4 after **Location** has been excluded—$\mathbf{Y}^4$.

| $(X_i : X_k)/l$ | 1 | 2 | 3 | $Y^4_{ik}(\Sigma)$ |
|---|---|---|---|---|
| $X_1 : X_2$ | 0 | 0 | 0 | 0 |
| $X_1 : X_3$ | 0 | 0 | 0 | 0 |
| $X_2 : X_3$ | 0 | 0 | 0 | 0 |

The equivalent exchange between $X_1$ and $X_2$ rows is carried out in Step 9. It consists of two simultaneous changes: the decrease in **Equipment** from 78 to 60 and the decrease in **Costs** from 1600 to 1200. The resultant preference matrix is shown in Table 14.

**Table 14.** The preference matrix at Iteration 5—$\mathbf{F}^5$.

| | 1: Equipment | 2: Size | 3: Costs |
|---|---|---|---|
| $X_1$: **Office A** | 60.0 | 700.0 | 1700.0 |
| $X_2$: **Office B** | 60.0 | 500.0 | 1200.0 |
| $X_3$: **Office C** | 50.0 | 850.0 | 1900.0 |

### 3.5. Iteration 5

The preference matrix at the fifth iteration $\mathbf{F}^5$ is presented in Table 14. The computed pairwise comparison matrix $\mathbf{Y}^5$ is shown in Table 15. The equivalent exchange between $X_1$ and $X_2$ rows is carried out at the fifth iteration. The increase in **Size** from 500 to 700 is equivalent to the increase in **Costs** from 1200 to 1500. The modified matrix after the fifth iteration is in Table 16.

**Table 15.** The pair-wise comparison matrix at Iteration 5—$\mathbf{Y}^5$.

| $(X_i : X_k)/l$ | 1 | 2 | 3 | $Y_{ik}^5(\Sigma)$ |
|---|---|---|---|---|
| $X_1 : X_2$ | 1 | 0 | 0 | 1 |
| $X_1 : X_3$ | 0 | 0 | 0 | 0 |
| $X_2 : X_3$ | 0 | 0 | 0 | 0 |

**Table 16.** The preference matrix at Iteration 6—$\mathbf{F}^6$.

| | 1: Equipment | 2: Size | 3: Costs |
|---|---|---|---|
| $X_1$: **Office A** | 60.0 | 700.0 | 1700.0 |
| $X_2$: **Office B** | 60.0 | 700.0 | 1500.0 |
| $X_3$: **Office C** | 50.0 | 850.0 | 1900.0 |

### 3.6. Iteration 6

Table 16 contains the preference matrix $\mathbf{F}^6$ at the beginning of the sixth iteration. The corresponding matrix $\mathbf{Y}^6$ of pairwise comparisons is shown in Table 17. The first row of the matrix in Table 17 contains all unity elements except one. This is a clear sign that a strictly dominated alternative has to be detected in Step 3. It happens to be $X_1$ 'Office A', which is inferior to $X_2$ 'Office B' by the singe attribute named **Cost**, while the estimates for two other attributes are equal for the alternatives in the pair $(X_1, X_2)$. The shrunk matrix with renumbered alternatives is shown in Table 18. The matrix containing the result of pairwise comparison is depicted in Table 19.

**Table 17.** The pair-wise comparison matrix at Iteration 6—$\mathbf{Y}^6$.

| $(X_i : X_k)/l$ | 1 | 2 | 3 | $Y_{ik}^6(\Sigma)$ |
|---|---|---|---|---|
| $X_1 : X_2$ | 1 | 1 | 0 | 2 |
| $X_1 : X_3$ | 0 | 0 | 0 | 0 |
| $X_2 : X_3$ | 0 | 0 | 0 | 0 |

**Table 18.** The preference matrix at Iteration 6 after 'Office A' has been excluded—$\mathbf{F}^6$.

| | 1: Equipment | 2: Size | 3: Costs |
|---|---|---|---|
| $X_1$: **Office B** | 60.0 | 700.0 | 1500.0 |
| $X_2$: **Office C** | 50.0 | 850.0 | 1900.0 |

**Table 19.** The pair-wise comparison matrix at Iteration 6 after 'Office A' has been excluded—$\mathbf{Y}^6$.

| $(X_i : X_k)/l$ | 1 | 2 | 3 | $Y_{ik}^6(\Sigma)$ |
|---|---|---|---|---|
| $X_1 : X_2$ | 0 | 0 | 0 | 0 |

The equivalent exchange is performed between two rows: the increase in **Equipment** from 50 to 60 is treated as equivalent to the increase in **Costs** from 1900 to 2000. It provides the modified preference matrix shown in Table 20.

**Table 20.** The preference matrix at Iteration 7—$\mathbf{F}^7$.

|  | 1: Equipment | 2: Size | 3: Costs |
|---|---|---|---|
| $X_1$: **Office B** | 60.0 | 700.0 | 1500.0 |
| $X_2$: **Office C** | 60.0 | 850.0 | 2000.0 |

*3.7. Iteration 7*

The preference matrix $\mathbf{F}^7$ at the beginning of the seventh iteration is drawn in Table 20. The computed matrix $\mathbf{Y}^7$ containing the pairwise comparisons is shown in Table 21.

**Table 21.** The pair-wise comparison matrix at Iteration 7—$\mathbf{Y}^7$.

| $(X_i : X_k)/l$ | 1 | 2 | 3 | $Y^7_{ik}(\Sigma)$ |
|---|---|---|---|---|
| $X_1 : X_2$ | 1 | 0 | 0 | 1 |

The attribute marked as **Equipment** takes the same value over both alternatives, so it appears to be insignificant and must be excluded. The shrunk preference matrix is shown in Table 22, and the matrix of pair-wise comparisons is presented in Table 23. Then, the equivalent exchange is being conducted where the decrease in **Size** from 850 to 700 is assumed to be equivalent to the decrease in **Costs** from 2000 to 1650. This exchange leads to the preference matrix depicted in Table 24.

**Table 22.** The preference matrix at Iteration 7 after 'Equipment' has been excluded—$\mathbf{F}^7$.

|  | 1: Size | 2: Costs |
|---|---|---|
| $X_1$: **Office B** | 700.0 | 1500.0 |
| $X_2$: **Office C** | 850.0 | 2000.0 |

**Table 23.** The pair-wise comparison matrix at Iteration 7 after 'Equipment' has been excluded—$\mathbf{Y}^7$.

| $(X_i : X_k)/l$ | 1 | 2 | $Y^7_{ik}(\Sigma)$ |
|---|---|---|---|
| $X_1 : X_2$ | 0 | 0 | 0 |

*3.8. Iteration 8*

Table 24 contains the preference matrix available at the beginning of Iteration 8. The accompanying comparison matrix is given in Table 25.

**Table 24.** The preference matrix at Iteration 8—$\mathbf{F}^8$.

|  | 1: Size | 2: Costs |
|---|---|---|
| $X_1$: **Office B** | 700.0 | 1500.0 |
| $X_2$: **Office C** | 700.0 | 1650.0 |

**Table 25.** The pair-wise comparison matrix at Iteration 8—$\mathbf{Y}^8$.

| $(X_i : X_k)/l$ | 1 | 2 | $Y^8_{ik}(\Sigma)$ |
|---|---|---|---|
| $X_1 : X_2$ | 1 | 0 | 1 |

The attribute named **Size** becomes insignificant and has to be put out of the consideration. The resultant matrix containing the rest column is shown in Table 26.

**Table 26.** The preference matrix at the end of Iteration 8—$\mathbf{F}^8$.

|  | 1: Costs |
| --- | --- |
| $X_1$: **Office B** | 1500.0 |
| $X_2$: **Office C** | 1650.0 |

*3.9. The Final Decision*

The resultant preference matrix in Table 26 allows for performing a simple comparison between all the available alternatives—there are only two rest, using the single scalar parameter **Costs**. Thus, 'Office B' is chosen as a unique optimal solution, since this choice implies a lower amount of money spent on renting the office.

**4. Discussion**

In the current section, the most important and controversial points regarding the proposed EEM are discussed. The list includes the description of the attributes, the foundation of EEM, the main strategy, the multiple output and the algorithm convergence issue.

*4.1. The Description of the Attributes*

Even though the attribute may not be take on numerical values interpreted as quantities, the order, at least a partial one [51], should exist. In other words, the attributes are expected to be measured in scales allowing comparison between their elements. Nonetheless, in almost all cases, the estimate of the attributes are stored in the form of scalars whose numerical values may be interpreted differently depending on what they actually represent. For instance, they can be natural numbers representing the ranks of the alternative among others with respect to some attribute. As an alternative example, they can be financial expenses or profits taking on their values out of the real number field, possibly with some additional restrictions; thus, expenses can only be a non-negative real number.

The complete set of the attributes can be divided into two groups which can be called *benefits* and *penalties*. The attributes belonging to the first group, the benefits, exhibit an ascending preference order: the larger the numerical value of the estimate is, the more preferable its value is. In contrast, the attributes belonging to the second group, the penalties, exhibit a descending preference order: the larger the numerical value of the estimate is, the less preferable its values are.

The DM should always bear in mind which of the groups the attributes belong to when they are about to enter the new value for the attribute engaged in the equivalent exchange. Thus, if the two attributes are both benefits, then an increase in the value of one attribute has to cause a decrease in another. This may look as an exchange of an amount of some goods for another amount of some other goods. The similar alterations may be carried out in the estimates of two attributes if they both are penalties, yet the reasoning will be a bit different there. The DM is ready to take up some additional amount of some inconvenience if another amount of some other inconvenience is lifted. However, if two attributes to be exchanged belong to different groups, an increase in one of them should also cause an increase in another, and vice versa for decreases. In other words, the DM is ready to purchase some additional amounts of goods with taking some extra amount of inconvenience, and vice versa for decreases.

*4.2. The Foundation of EEM*

The key idea lying behind EEM finds its foundation in the psychological aspect of the problems involving number comparisons [52]. It looks evident that if there is only one attribute to estimate the alternatives, and there will be no difficulty to say which of them is more preferable. However, when it comes to the comparison where multiple criteria are involved, starting with three, the personal perception of the complexity becomes significantly higher. There are several possible approaches to overcoming this complexity.

The first and possibly the simplest one is the introduction of the universal merit, which can be either one of the attributes, typically the one measured in currency, or some auxiliary metric. Then, exchange rates are established, which allows the DM to find the equivalent value in the merit. However, there are at least two disadvantages here. The first one is that the exchange rate remains linear while the marginal value of the attribute may well depend on its current value. The second one is more fundamental: due to their meanings, the attributes can form a so-called incomparable set. This typically happens when there are ordinal, true scalar, and relative scalars in the set of attributes. To make things worse, their estimates can be found by experts, and that fact will inevitably bring in uncertainty.

The second solution, more advanced, could be defining the target function, which can look like a stricter mathematical formalization of the preference problem. Although it can cope with the nonlinear exchange rates, the amount of effort and knowledge in the problem area required to define this function accurately turns out to be dramatically high, especially in the case of more than three attributes. However, the set of the alternatives is always finite so that the optimization problem remains rather sparse and that effort may look wasteful in the end. The combination of incomparable attributes can make the definition of the target function rather laborious. To make things worse, if the ordinals are among the attributes, the target function ceases to be continuous. That will dramatically increase the complexity of the mathematical tools required for solving the optimization problem.

The third approach, which EEM is actually based upon, consists of transforming the initial search in the space of alternatives being compared in the sequence of simple numerical comparisons where only two attributes are involved. Thus, the DM is expected to be able to make each exchange where they have to find the reasonable equivalence between changes in one and another attributes. Therefore, the comparison at each step where the expert's knowledge is required can be carried out in the conditions which remain plausible for a general human being's perception of a simple trade-off.

### 4.3. The Main Strategy

The main outline of the algorithm, which is shown in Figure 1, as well as its iterations observed in Section 3, could make an impression that the main goal is an iterative elimination of the attributes and deletion of the alternatives. Indeed, an attribute can be deleted as soon as it can be considered as an insignificant one, while an alternative can be deleted from the list if it is a dominated one. In other words, there exists another alternative which is dominating over one to be deleted. Consequently, the DM can think to follow two main strategies:

1. The first strategy consists of conducting such a sequence of exchanges which results in the equal values for the same attribute which the estimates take on in all the alternatives.
2. The second strategy consists of the equalization of two alternatives by all parameters except one; the only parameter with different values will determine the dominating alternative.

The second strategy looks more advantageous since it leads to crossing out less preferable alternatives, which is generally speaking the desirable result. However, the elimination of the attributes reduces the overall dimensionality of the problem. This makes the DM's perception more clear at further iterations. Alternatively, if the DM applies EEM as an auxiliary tool only, they can stop the algorithm run after some iteration when the preference matrix becomes small enough and free of onerous attributes, so the DM can decide to switch to another methods afterwards.

### 4.4. The Multiple Algorithm Outputs

Since the number of the alternatives to make a comparison between tends to reduce with every iteration as well as the number of the attributes involved in the comparison, only a few alternatives will eventually remain. Hence, the dominating alternatives chosen at the final iteration of the algorithm will become the output of the whole algorithm.

The result of the algorithm is usually a single alternative—$X^*$. In cases where all attributes except one have been excluded, the algorithm may produce several alternatives at the final iteration. It may simply happen if the estimates for such alternatives being compared by the single attribute turn out to be equal. Those alternatives will generally be thought of as if they have equal merit as the preferable options for the DM. If such an output as multiple alternatives is not acceptable, the DM has to pick one of them at random.

### 4.5. The Convergence of the Algorithm

Convergence of the algorithm arises from the fact that actions carried out in steps 7, 8, and 9 ensure the decrease in value $M^q$ (12). It follows from the fact that the number of attributes, by which alternatives $(\hat{i}, \hat{k})$ differ, will decrease by 1. In other words, upon transition to step $q + 1$, the inequality $M^{q+1} < M^q$ is guaranteed. This means that inequality $M^{q+1} < 2$ will be fulfilled in a certain step, which means that a dominated alternative is present and, therefore, is bound to be excluded according to the action done in step 4. This way, the total number of alternatives and attributes will consistently be reducing so that the preference matrix will gradually become shrunk. This, in turn, ensures that the conditions for the start of the algorithm's final steps 5 or 6 will be inevitably met.

### 4.6. The Multiple Attribute Dominance

The base version of the algorithm, which is described in Section 2, has been deliberately designed so as to identify the dominated alternative among two in the case where there is a difference in the estimate by a single attribute only. This can be seen as the algorithm drawback to some extent since a dominated alternative may have been found in some cases where the difference in the estimates by more than one attribute takes place. The simple example involving two attributes for the sake of simplicity is briefly investigated below.

Suppose that, for two alternatives, namely $X_i$ and $X_k$, there are two attributes indexed with $l_1$ and $l_2$ such that $F_{il_1} \succ F_{kl_1}$ AND $F_{il_2} \succ F_{kl_2}$, while their estimates measured for every other attributes are equal to each other. It appears evident for the DM that $X_i$ alternative dominates over $X_k$. Hence, the latter is ready to be called dominated and then excluded from the list of alternatives in Step 4. However, the current version of EEM algorithm does not identify this case especially and, therefore, at Step 3, this will lead to Case C but not Case B. Indeed, since the two alternatives differ in two attributes, it is guaranteed that this pair of attributes $(l_1, l_2)$ will be chosen at Steps 7 and 8 as the attributes for the exchange: $\hat{l}$ and $l^*$ at the current or a subsequent iteration. Then, the equivalent exchange at Step 9 will make estimates equal for $\hat{l}$, whereas the difference in $l^*$ is built up even bigger than it was before the exchange. Thus, the order is preserved between $X_i$ and $X_k$ and $X_k$ goes on being dominated. The alternative will eventually be identified as a dominated one at a subsequent iteration as soon as only one attribute with different values remains.

Nonetheless, the reliable immediate identification of strict dominance in cases where the alternatives differ in estimates by more than one attribute simultaneously is expected to bring about a shortcut in the algorithm run. This will help to decrease the total number of required iterations and the effort the DM is bound to make during the equivalent exchange steps. This is obliged to be one of the desirable improvements for the next version of EEM and its algorithmic implementation.

## 5. Conclusions

The main result achieved in the research presented in the paper is the formalization of the equivalent exchange method in the form of the algorithm ready for computer-assisted execution. The applicability of the EEM algorithm as a tool for making decisions has been clearly demonstrated by an example where the alternatives were estimated by the list of incomparable attributes. The simulation showed the full run of the algorithm, from the initial settings of preference matrix to the final iteration where the single alternative would be found.

The current version of EEM presented in the paper determines the whole algorithm as mostly an expert-oriented or expert-driven tool. It means that the algorithm offers the DM a lot of freedom for choosing the attributes and alternatives to conduct the equivalent exchange, yet the EDSS can suggest the most appropriate attribute for exchange. Furthermore, it is the DM's entire privilege to determine the changes in the values of the estimates by the attributes engaged in the exchange. It basically demands a relatively high level of the DM's qualifications and their unbiasedness and reasonable neutrality in making decisions about exchange at each iteration. However, the pair-wise manner of comparison can help to diminish the DM's personal preference for some alternatives if it happens to exist.

Another feature of EEM is that the solution it generates crucially depends on DM's personal choices. In other words, it cannot be guaranteed that the output alternatives obtained by two different DM will be the same. This may be thought as the disadvantage of EEM to some extent. However, this may bring the ability for coordinating positions of two DM. If they both have come to the same result, it will make a strong point for agreement. Otherwise, if the results turn out to be different, it allows for reducing the number of possible alternatives which will be compared further.

A possible way of making the exchange more systematic can be considered if pairwise objective functions have been properly introduced. Despite the fact these functions can not be defined plausibly for all possible pairs of attributes, they may help DMs to evaluate the new value entered for the attribute to be changed. However, the selection of the most appropriate structure of each function will inevitably require a large amount of expert knowledge, while a more general problem—the invention of the methodology for computer-aided building these functions—is another challenging task, since those functions do not appear to be linear in a wide range of values the variable take on.

The additional result achieved in this study was the implementation of the newly developed method into the EDSS. This goal has been successfully achieved since the modular architecture of the server part of the system allows for adding new decision-making techniques without changing the source code of the main system. Thus, the user graphical interface of the EDSS was updated, the proposed method was implemented as a program module written in C# language, and the knowledge base exploited by the EDSS was expanded. Unfortunately, the current version of the EDSS cannot be made available to a wide audience due to existing limitations in its license. However, the work on its newer version which can be distributed by some open source license is going on.

The further development in the practical implementations of the algorithm will be focused mostly on improving the usability by extending the graphical user interface supporting the DM. For instance, the module may provide an inexperienced user with more verbose tips about the process and the decisions the DM is about to make at each step. Additionally, the current module can be extended with some heuristic procedures which can suggest to the DM those attributes and alternatives whose changes are expected to be relatively smaller, in absolute or relative values, with respect to others.

**Author Contributions:** Conceptualization, T.K.; methodology, T.K.; software, T.K.; validation, T.K., T.S.; formal analysis, T.S.; investigation, T.S.; resources, T.S.; data curation, T.K.; writing—original draft preparation, T.K. and T.S.; writing—review and editing, T.K. and T.S.; visualization, T.K.; supervision, T.K.; project administration, T.S.; funding acquisition, T.S. All authors have read and agreed to the published version of the manuscript.

**Funding:** This research was funded by the state assignment of the Ministry of Science and Higher Education of the Russian Federation, project No. FSFF-2020-0015.

**Institutional Review Board Statement:** Not applicable

**Informed Consent Statement:** Not applicable

**Data Availability Statement:** Not applicable

**Conflicts of Interest:** The authors declare no conflict of interest.

**Abbreviations**

The following abbreviations are used in this manuscript:

| | |
|---|---|
| AIM | Aspiration-level Interactive Method |
| DM | Decision Maker |
| DSS | Decision Support System |
| EDSS | Expert Decision Support System |
| EEM | Equivalent Exchange Method |
| ELECTRE | ÉLimination Et Choix Traduisant la REalité |
| | (Elimination and Choice Translating Reality) |
| GUEST | Acronym for the methodology: Go, Uniform, Evaluate, Solve, Test |
| PROMETHEE | Preference Ranking Organization METHod for Enrichment of Evaluations |
| SMAA | Stochastic Multiobjective Acceptability Analysis |

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
