# Peer review of "Equivalent Exchange Method for Decision-Making in Case of Alternatives with Incomparable Attributes"

_inventions, doi:10.3390/inventions8010012_

Round 1

Reviewer 1 Report

The paper presented us with an extremely useful automated decision-making system. However, it requires more explanations and a bibliography that shows the novelty and the way of implementation, describing more clearly the decisions. Please also specify the disadvantages of using the method as well as ways and work needed to improve it.

Author Response

In the attached file you could find our response to your review and the paper with all corrections we made during the revision.

Reviewer 2 Report

Dear authors, 

the paper is interesting and tackles a very important problem. The following is a list of suggestions to improve the quality of the paper:

- "Aspiration-level interactive method (AIM)" -> "Aspiration-level Interactive Method (AIM)"

- a recent paper describing an application of a DSS for the supply car industry is: 10.3390/su14042408

- An important technique dealing with the problem of a DM which is not really able to decide the right objective function and/or model, has been tackled in the context of the location of electric vehicles charging stations in: 10.15439/2019F171, 10.1007/978-3-030-58884-7_4, 10.1016/j.tre.2020.102174

- Page 3 lines 130 on: add a bulletpoint list. 

- Page 5 line 179: "is being built" -> "is built"

- Line 305 page 9, there is something wrong, as well as in lines 301, 293, ect.

- sentence: "The EDSS was designed in order to simplify making necessary calculations for substantiation of choosing among alternatives by means of verified formal mathematical methods based on models commonly adopted in the field of expert estimations in economics." Is not much clear, please revise it.

- I would not say that "Practically, the proposed method has been implemented as a program module written in C# language with support program interface providing the data structure interchange at the level enough to the full inclusion into the system."  Just in the conclusion. It would be nice to highlight it before (e.g., in the simulation example). Moreover, what about sharing the code on some free platform? It could be beneficial for future studies as well as letting reviewers to better check your algorithm.

- Lines 510-521 can be better explained in a technical section (maybe with the above point) in which you point out the general IT structure that you have implemented.

- In the text, you say "most effective solution" (see e.g. line 361). Do you mean non dominanted solutions? Or solutions belonging to the Pareto front?

Author Response

(The authors gave the same response as above.)

Reviewer 3 Report

The paper presents a new decision-making technique based on the equivalent exchange. The method proposes an algorithm capable to select the best option among a set of alternatives by reducing progressively the number of attributes or alternatives. The paper is not completely well structured but it is well written. Although the content can be considered relevant to the topic of decision-making, several major issues shall be addressed before acceptance for publication:

1.       Generally speaking, there is a lack in the mathematical characterisation of the concept of "equivalence". Without a proper definition of the laws governing the equivalence between couples of attributes, there is no guarantee that the DM apply always the same process in exchanges that he makes at different steps between the same attributes considering different alternatives. The concept of equivalence shall be better defined and discussed.

2.       Section 2.2. The process described in step 3 is not consistent with the definition of dominance given on page 4. In detail, it is completely wrong to assert that in case C there are no dominated alternatives. In fact, an alternative might perform not better than another in all the attributes, although more than one attribute value is different (so even if Y^q_ik < L^q -1). Such a situation can be easily tackled by classifying the attributes as cost or benefit attributes in order to identify all dominated options.

3.       It is the opinion of the reviewer that the proposed process leaves too much freedom to the DM. Often situations in which the choice of the attributes to carry out the equivalent exchange is arbitrary. This is evident in the worked example in which often there are many possible choices but one is preferred without any explanation. How can be assured that by choosing a couple of attributes instead of another, the same alternative will be finally selected? A discussion of this topic is required.

4.       The excessive reliance on the DM has another drawback: the proposed method can be scarcely automated. This topic shall be discussed and possibly a solution given. Maybe, defining mathematical laws for equivalence and sorting the attributes by priority to carry out equivalent exchanges in a guided way, when multiple alternatives are available, could be considered as a solution.

5.       The worked example and the process that led to choosing the attributes for exchange shall be better described.

6.       A discussion section is lacking.

7.       The conclusions shall be totally rewritten. Currently, they provide the context of the work, which is more suitable for the introduction or a dedicated section of the paper. The conclusions shall summarise the most important findings of the work, they shall describe how the research question stated in the introduction has been answered and finally give perspectives.

8.       there are several misprints in the paper (for instance the Chinese character at lines 293, 301, 305, etc.). Please revise carefully the paper.

It is the opinion of the reviewer that the paper shall be reconsidered after major revision.

Author Response

(The authors gave the same response as above.)

Round 2

Reviewer 1 Report

Dear Authors, You did a great work and your paper is ready to be published, in my opinion.

Best wishes

Author Response

We are thankful to you for your comments which helped us significantly improve the quality of our paper.

Reviewer 2 Report

Dear authors, 

I think that the paper is now ready to be published. 

Best Regards

Author Response

(The authors gave the same response as above.)

Reviewer 3 Report

Most of the flaws have been well addressed by the authors or acknowledged in conclusions. Now the paper is more readable, clear and consistent. There is only one remaining issue:

Line 260-261 “there are no dominated alternatives”: this statement is false as already pointed out. For instance, one alternative can differ from another by two attributes (Y^q_ik = 2), but, if one alternative is better than the other in both different attributes, the first is still dominating the second one. Hence, in Case C, dominated alternatives can still be included while they should be removed. This is why it was suggested to distinguish between cost and benefit attributes since the beginning: this can ease the complete application of filtering by dominance.

If this issue is fixed, the paper can be published.

Author Response

We highly appreciate your genuine interest in our research and we are thankful to you for your comments which helped us significantly improve the quality of our paper.

Authors’ reply to the reviewer’s comment is following:

1. Line 260-261 “there are no dominated alternatives”: this statement is false as already pointed out. For instance, one alternative can differ from another by two attributes (Y^q_ik = 2), but, if one alternative is better than the other in both different attributes, the first is still dominating the second one. Hence, in Case C, dominated alternatives can still be included while they should be removed. This is why it was suggested to distinguish between cost and benefit attributes since the beginning: this can ease the complete application of filtering by dominance.

We basically agree with you point. In order to address this point more delicate we have added a short remark in the lines where we describe Step 3 and a much bigger explanation as a new subsection in the Discussion.

The comparative version is in the attached file.

The remark:

Case C is the situation where neither Case A, nor Case B take place. Then, there are no dominated alternatives which have been identified in the step.

It is worth noting that Case C does not imply that there is no dominated alternatives

in the set of the alternatives. However, the searching procedure relying on the difference in a single attribute cannot identify it immediately. The more details about this point are discussed in Subsection 4.6. 

The longer explanation (Subsection 4.6):

The multiple attribute dominance

The base version of the algorithm, which is described in Section 2, has been deliberately designed so as to identify the dominated alternative among two in case where there is a difference in the estimate by a single attribute only. This can be seen as the algorithm drawback to some extent since a dominated alternative may have been found in some cases where the difference in the estimates by more than one attribute takes place. The simple example involving two attributes for the sake of simplicity is briefly investigated below.

Suppose that for two alternatives, namely Xi and Xk, there are two attributes indexed with l1 and l2 such that Fil1 ≻ Fkl1 AND Fil2 ≻ Fkl2, while their estimates measured for every other attributes are equal to each other. It appears evident for the DM that Xi alternative dominates over Xk. Hence, the latter is ready to be called dominated and then excluded from the list of alternatives in Step 4. However, the current version of EEM algorithm does not identify this case specially and, therefore, at Step 3, this will lead to Case C but not Case B. Indeed, since the two alternatives differ in two attributes, it is guaranteed that this pair of attributes l1, l2 will be chosen at Step 7 and 8 as the attributes for the exchange: \hat{l} and l* at the current or a subsequent iteration. Then, the equivalent exchange at Step 9 will make estimates equal for \hat{l} whereas the difference in l* is built up even bigger than it was before the exchange. Thus, the order is preserved between Xi and Xk and Xk goes on being dominated. The alternative will eventually be identified as a dominated one at a subsequent iteration as soon as only one attribute with different values remains.

Nonetheless, the reliable immediate identification of strict dominance in cases where the alternatives differ in estimates by more than one attribute simultaneously is expected to bring about a shortcut in the algorithm run. This will help to decrease the total number of required iterations and the effort the DM is bound to make during the equivalent exchange steps. This is obliged to be one of the desirable improvements for the next version of EEM and its algorithmic implementation.
